Splice site m6A methylation prevents binding of DGCR8 to suppress KRT4 pre-mRNA splicing in oral squamous cell carcinoma

Li Xiaoxu
Fang Juan
Tao Xiaoan
Xia Juan
Cheng Bin chengbin@mail.sysu.edu.cn
Wang Yun wangyun23@mail.sysu.edu.cn
Guangdong Provincial Key Laboratory of Stomatology, Guanghua School of Stomatology, Hospital of Stomatology, Sun Yat-sen University , Guangzhou , Guangdong , China
Tyagi Abhishek
Electronic publication date: 2023 Feb 16
Publication date: 2023
Volume: 11
Electronic Location ID: e14824
Received 2022 Oct 27; Accepted 2023 Jan 8
Copyright: ©2023 Li et al.
Copyright year: 2023
Copyright holder: Li et al.
License: This is an open access article distributed under the terms of the Creative Commons Attribution License, which permits unrestricted use, distribution, reproduction and adaptation in any medium and for any purpose provided that it is properly attributed. For attribution, the original author(s), title, publication source (PeerJ) and either DOI or URL of the article must be cited.
License URL: https://creativecommons.org/licenses/by/4.0/

Keywords: Oral squamous cell carcinoma, Keratin 4, m6A methylation, Pre-mRNA splicing

Funding: National Natural Science Foundation of China 81500864 Guangdong Basic and Applied Basic Research Foundation 2019A1515011203 This work was supported by the National Natural Science Foundation of China (Grant No. 81500864) and Guangdong Basic and Applied Basic Research Foundation (Grant No. 2019A1515011203). The funders had no role in study design, data collection and analysis, decision to publish, or preparation of the manuscript.

==============================
Oral squamous cell carcinoma (OSCC) is the 11th most prevalent tumor worldwide. Despite advantages of therapeutic approaches, the 5-year survival rate of patients with OSCC is less than 50%. It is urgent to elucidate mechanisms underlying OSCC progression for developing novel treatment strategies. Our recent study has revealed that Keratin 4 (KRT4) suppresses OSCC development, which is downregulated in OSCC. Nevertheless, the mechanism downregulating KRT4 in OSCC remains unknown. In this study, touchdown PCR was utilized to detect KRT4 pre-mRNA splicing, while m6A RNA methylation was identified by methylated RNA immunoprecipitation (MeRIP). Besides, RNA immunoprecipitation (RIP) was used to determine RNA-protein interaction. Herein, this study indicated that intron splicing of KRT4 pre-mRNA was suppressed in OSCC. Mechanistically, m6A methylation of exon-intron boundaries prevented intron splicing of KRT4 pre-mRNA in OSCC. Besides, m6A methylation suppressed the binding of splice factor DGCR8 microprocessor complex subunit (DGCR8) to exon-intron boundaries in KRT4 pre-mRNA to prohibit intron splicing of KRT4 pre-mRNA in OSCC. These findings revealed the mechanism downregulating KRT4 in OSCC and provided potential therapeutic targets for OSCC.

Introduction

Oral squamous cell carcinoma (OSCC) is the 11th most prevalent tumor worldwide, which accounts for 2–3% of all cancers (Ferlay et al., 2010; Hasegawa et al., 2021). It is estimated that there are around 350,000 new cases and 170,000 deaths from OSCC annually (Bray et al., 2018; Yang et al., 2021). Despite advantages of therapeutic approaches, the 5-year survival rate of patients with OSCC is less than 50% due to high rates of recurrence and metastasis (Bloebaum et al., 2014; Panzarella et al., 2014). Therefore, it is essential to elucidate mechanisms underlying OSCC progression for developing novel treatment strategies.

Keratin 4 (KRT4) is a member of the type II keratin family (Zhang et al., 2018). Previous studies have indicated that KRT4 expression is downregulated in OSCC (Lallemant et al., 2009; Toruner et al., 2004; Ye et al., 2008). Our recent study has also revealed that KRT4 expression is decreased in OSCC cells and KRT4 suppresses autophagy related 4B cysteine peptidase (ATG4B)-mediated autophagy to inhibit OSCC development (Li et al., 2022). Nevertheless, the mechanism downregulating KRT4 mRNA in OSCC remains unknown.

Dysregulation of pre-mRNA splicing would lead to the downregulation of mRNA (Han et al., 2011; Lee & Rio, 2015). Growing evidence has demonstrated that RNA methylation contributes to proper pre-mRNA splicing and subsequent mRNA expression. For instance, N6-methyladenosine (m6A) reader YTH domain containing 1 (YTHDC1) associates with m6A modified pre-mRNA and facilitates exon inclusion during pre-mRNA splicing by recruiting serine and arginine rich splicing factor 3 (SRSF3) whereas blocking SRSF mRNA binding (Xiao et al., 2016). In addition, the m6A methylation of 3′ spice site in S-adenosylmethionine (SAM) synthetase pre-mRNA prevents RNA splicing through inhibiting binding of splicing factor U2 small nuclear RNA auxiliary factor 1 (U2AF1) to the 3′ spice site (Mendel et al., 2021). Yet the role of m6A methylation in KRT4 pre-mRNA splicing has not been reported.

DGCR8 microprocessor complex subunit (DGCR8) is an essential splicing factor for microRNA (miRNA) processing (Guo & Wang, 2019; Michlewski & Caceres, 2019). Besides, DGCR8 is also involved in RNA methylation-mediated miRNA processing. A previous study has indicated that m6A methylation enhances the binding of DGCR8 and primary miR-19a to facilitate miRNA processing in cardiovascular endothelial cell (Zhang et al., 2020a). Except regulating miRNA splicing, DGCR8 contributes to mRNA processing as well. In mouse embryonic stem cells, DGCR8 interacts with transcription factor 7 like 1 (Tcf7l1) pre-mRNA to promote the splicing of Tcf7l1 pre-mRNA (Cirera-Salinas et al., 2017). However, the role of DGCR8 in RNA methylation-mediated KRT4 pre-mRNA splicing is largely unknown.

Therefore, the primary aim of the current study was to investigate the effects of m6A methylation and DGCR8 on KRT4 pre-mRNA splicing in OSCC.

Materials and Methods

Cell culture

Normal oral keratinocytes (NOK) and OSCC cell line HN6 cells were obtained from Cell Bank at the Chinese Academy of Sciences (Shanghai, China) and cultured as previously described (Li et al., 2022).

Cell transfection

Cells were transfected with METTL3, METTL14 or DGCR8 siRNA and siRNA negative control (NC) by Lipofectamine 2000 (Invitrogen, Carlsbad, CA, USA) as previously described (Li et al., 2022). Then cells were collected for following experiments at 48 h after transfection. Sequences of siRNAs were listed in Table 1.

Table 1 Sequences of siRNA used in this study.

SiRNA		Sequences (5′–3′)	
METTL3 siRNA	Sense	CACAGAGTGTCGGAGGTGATTC	
	Antisense	CTGTAGTACGGGTATGTTGAGCC	
METTL14 siRNA	Sense	GACCTTGGAAGAGTGTGTTTACG	
	Antisense	CTTTGATCCCCATGAGGCAGT	
DGCR8 siRNA	Sense	ACAUCUUGGGCUUCUUUCGAG	
	Antisense	CGAAAGAAGCCCAAGAUGUCC	
siRNA NC	Sense	UUCUCCGAACGUGUCACGU	
	Antisense	ACGUGACACGUUCGGAGAA	

Bioinformatics analysis

RMBase v2.0 database (https://rna.sysu.edu.cn/rmbase/) and RMVar database (https://rmvar.renlab.org/) were used to explore potential m6A modification sites in or nearby KRT4 pre-mRNA splicing sites. Besides, ENCORI database (https://starbase.sysu.edu.cn/index.php) was utilized to mine RNA-protein interactions.

Touchdown polymerase chain reaction (PCR)

Touchdown PCR was utilized to detect KRT4 pre-mRNA splicing. First, total RNA was extracted from cells by TRIzol reagent (Invitrogen) followed by cDNA synthesis using PrimeScript RT Reagent Kit (Takara, Dalian, Liaoning, China). Subsequently, touchdown PCR was performed in a volume of 50 µL using Phanta Max Buffer (Vazyme, Nanjing, Jiangsu, China) as follows: 95 °C for 3 min, 95 °C for 15 s, 74 °C for 90 s for 5 cycles; 95 °C for 15 s, 72 °C for 90 s for 5 cycles; 95 °C for 15 s, 70 °C for 90 s for 5 cycles; 95 °C for 15 s, 68 °C for 90 s for 25 cycles followed by 68 °C for 5min. Sequences of primers used for touchdown PCR were listed in Table 2.

Table 2 Sequences of primers for PCR used in this study.

Genes		Sequences (5′–3′)	
KRT4 pre-mRNA (E1/E2)	Forward	CTCCTCAACAACAAGTTTGCCTC	
	Reverse	CTTTGTCATTGCCCAAGGTATC	
KRT4 pre-mRNA (E2/E3)	Forward	AGCCCCTCTTTGAGACCTACC	
	Reverse	TCATTCTCGGCTGCTGTGC	
KRT4 pre-mRNA (E3/E4)	Forward	GCACAGCAGCCGAGAATGAC	
	Reverse	TGTTCAGGTAGGCAGCATCCAC	
KRT4 pre-mRNA (E4/E5)	Forward	GGATGCTGCCTACCTGAACAAG	
	Reverse	TTGGTCTGGTACAGGGCTTCAG	
KRT4 pre-mRNA (E5/E6)	Forward	CGAGGAGATTGCCCAGAGGA	
	Reverse	CAGCCTCTGGATCATCCTGTTG	
KRT4 pre-mRNA (E6/E7)	Forward	GATCTCGGTTGACCAACATGG	
	Reverse	TCCTGGTACTCACGCAGCATT	
KRT4 pre-mRNA (E7/E9)	Forward	AAGATGCCCACAGCAAGCG	
	Reverse	AGACACTGCCACCAAACCCA	

Quantitative real-time polymerase chain reaction (qRT-PCR)

After RNA extraction and cDNA synthesis, qRT-PCR was performed by the ABI7300 Real-Time PCR system (Applied Biosystems, Foster City, CA, USA) using TB Green® Premix Ex Taq™ II (Tli RNaseH Plus) (Takara) as previously described (Li et al., 2022). Sequences of primers used for qRT-PCR were listed in Table 3.

Table 3 Sequences of primers for qRT-PCR used in this study.

Genes		Sequences (5′–3′)	
KRT4 pre-mRNA (E2/E3)	Forward	AGCCCCTCTTTGAGACCTACC	
	Reverse	TCATTCTCGGCTGCTGTGC	
KRT4 pre-mRNA (E3/E4)	Forward	GCACAGCAGCCGAGAATGAC	
	Reverse	TGTTCAGGTAGGCAGCATCCAC	
KRT4 pre-mRNA (E5/E6)	Forward	CGAGGAGATTGCCCAGAGGA	
	Reverse	CAGCCTCTGGATCATCCTGTTG	
KRT4-E2(3)	Forward	TGCAACTAATTACGTGGATA	
	Reverse	TTCTTTAGGACCACAAAGTC	
KRT4-E3(4)	Forward	AGAGGAGATCAACAAACGCACAG	
	Reverse	AACCCATGACTTCAGCCAAAGA	
KRT4-E5(6)	Forward	CCAGATGCAGACCCATGTCAG	
	Reverse	GCTTGAGCTAATGATCACCTGTTC	
KRT4 mRNA	Forward	CATTGATCGCTGGGGTTGA	
	Reverse	ATACCCTTGACCGAAGACCG	
METTL3	Forward	CACAGAGTGTCGGAGGTGATTC	
	Reverse	CTGTAGTACGGGTATGTTGAGCC	
METTL14	Forward	GACCTTGGAAGAGTGTGTTTACG	
	Reverse	CTTTGATCCCCATGAGGCAGT	
DGCR8	Forward	CAAGATGCACCCACAAAGAAAG	
	Reverse	GATCCGTAAGTCACACCATCAA	
GAPDH	Forward	AACGGATTTGGTCGTATTGGG	
	Reverse	CCTGGAAGATGGTGATGGGAT	

Methylated RNA immunoprecipitation (MeRIP)

NOK and HN6 cells were collected and lysed. Then nucleic acid fragments were interrupted by ultrasound. Next, cell lysate was incubated with 1 µL m6A antibody (1:500, #ab208577, Abcam, Cambridge, MA, USA) overnight at 4 °C. Subsequently, m6A antibody and methylated RNA fragments were captured by avidin magnetic beads, and the level of methylated RNA was detected by qRT-PCR.

RNA immunoprecipitation (RIP)

RIP was performed by RNA-Binding Protein Immunoprecipitation Kit (Merck Millipore, Billerica, MA, USA). Briefly, HN6 cells were collected and lysed. Then nucleic acid fragments were interrupted by ultrasound. Subsequently, cell lysate was incubated with 1 µL METTL3 (#15073-1-AP, Proteintech; Rosemont, IL, USA), METTL14 (#26158-1-AP; Proteintech) or DGCR8 antibody (#60084-1-Ig; Proteintech) at 4 °C overnight. Next, protein-binding RNA fragments were captured by avidin magnetic beads, and the level of protein-binding RNA was identified by qRT-PCR.

Statistical analysis

Quantitative data of the current study were present as mean ±  standard deviation (SD) and statistical differences were analyzed by SPSS 20 software (SPSS Inc., Chicago, IL, USA) as previously described (Li et al., 2022). Besides, P < 0.05 was considered as statistically significant.

Results

Intron splicing of KRT4 pre-mRNA is suppressed in OSCC

To explore the mechanism downregulating KRT4 in OSCC, splicing of KRT4 pre-mRNA was detected by touchdown PCR. Compared to NOK cell, splicing of intron 2 between exon 2 and 3 (E2/E3), intron 3 between exon 3 and 4 (E3/E4), and intron 5 between exon 5 and 6 in KRT4 pre-mRNA (E5/E6) was inhibited in HN6 cells (Figs. 1A and 1B). Red arrows indicated fragments of KRT4 pre-mRNA containing exon-intron structures (Figs. 1A and 1B). Moreover, results of qRT-PCR confirmed that junctions of exon 2-exon 3 (E2/E3), exon 3-exon 4 (E3/E4) and exon 5-exon 6 (E5/E6) in KRT4 mRNA were dramatically decreased in HN6 cells compared to those in NOK cells, respectively (Fig. 1C). Therefore, these data suggested that intron splicing of KRT4 pre-mRNA was suppressed in OSCC.

Figure 1 Intron splicing of KRT4 pre-mRNA is suppressed in OSCC.

(A) Schematic diagram of KRT4 pre-mRNA. (B) Fragments of KRT4 pre-mRNA containing exon-intron structures or KRT4 mRNA detected by PCR in NOK cells and HN6 cells. Red arrows indicated fragments of KRT4 pre-mRNA containing exon-intron structures. (C) Levels of junctions of exon 2-exon 3 (E2/E3), exon 3-exon 4 (E3/E4) and exon 5-exon 6 (E5/E6) in KRT4 mRNA detected by qRT-PCR in NOK cells and HN6 cells. E, exon. ∗∗∗P < 0.001, ∗∗∗∗P < 0.0001.

m6A levels of exon-intron boundaries in KRT4 pre-mRNA is increased in OSCC

Bioinformatics analysis showed that there were m6A sites in exon 3 (KRT4-E3(4)) and exon 5 (KRT4-E5(6)) of KRT4 pre-mRNA nearby exon-intron boundaries (Fig. 2A). Besides, results of MeRIP found that m6A levels of m6A sites in exon 3 (KRT4-E3(4)) and exon 5 (KRT4-E5(6)) of KRT4 pre-mRNA were significantly elevated in HN6 cell compared to those in NOK cells (Figs. 2B and 2C). Thus, m6A methylation might be involved in the regulation of KRT4 pre-mRNA splicing in OSCC.

Figure 2 m6A levels of exon-intron boundaries in KRT4 pre-mRNA is increased in OSCC.

(A) Potential m6A modification sites in or nearby KRT4 pre-mRNA splicing sites. (B) The m6A level of m6A sites in exon 3 (KRT4-E3(4)) of KRT4 pre-mRNA detected by MeRIP. (C) The m6A level of m6A sites in exon 5 (KRT4-E5(6)) of KRT4 pre-mRNA detected by MeRIP. E, exon. ∗∗P < 0.01.

m6A methylation of exon-intron boundaries prevents intron splicing of KRT4 pre-mRNA in OSCC

Results of RIP performed by METTL3 and METTL 14 antibodies showed that m6A writers METTL3 and METTL14 associated with m6A sites in exon 3 (KRT4 pre-mRNA-E3(4)) and exon 5 (KRT4 pre-mRNA-E5(6)) of KRT4 pre-mRNA nearby exon-intron boundaries in HN6 cells (Fig. 3A), suggesting that METTL3 and METTL14 should modify m6A levels of m6A sites in exon 3 (KRT4 pre-mRNA -E3(4)) and exon 5 (KRT4 pre-mRNA -E5(6)) of KRT4 pre-mRNA in HN6 cells.

Figure 3 m6A methylation of exon-intron boundaries prevents intron splicing of KRT4 pre-mRNA in OSCC.

(A) Quantification of KRT4 pre-mRNA containing exon-intron structures by qRT-PCR following RIP performed by METTl3 or METTL14 antibody in HN6 cells. (B) The m6A level of m6A sites in exon 3 (KRT4-E3(4)) and exon 5 (KRT4-E5(6)) of KRT4 pre-mRNA detected by MeRIP in HN6 cells. (C) Levels of junctions of intron 2-exon 3 (KRT4-E(2)3), exon 3-intron 3 (KRT4-E(3)4) and exon 5-intron 5 (KRT4-E(5)6) in KRT4 pre-mRNA detected by qRT-PCR in HN6 cells treated with or without METTL3 and METTL14 siRNA. (D) Levels of KRT4 mRNA in HN6 cells treated with or without METTL3 and METTL14 siRNA. NC, negative control; siMETTL3, METTL3 siRNA; siMETTL14, METTL14 siRNA; E, exon. ∗∗P < 0.01, ∗∗∗P < 0.001, ∗∗∗∗P < 0.0001.

Next, METTL3 and METTL14 were silenced by siRNAs in HN6 cells (Fig. S1) to identify the effect of m6A methylation on intron splicing of KRT4 pre-mRNA in OSCC. Silence of METTL3 and METTL14 decreased m6A levels of m6A sites in exon 3 (KRT4 pre-mRNA -E3(4)) and exon 5 (KRT4 pre-mRNA -E5(6)) of KRT4 pre-mRNA in HN6 cells (Fig. 3B). Then intron splicing of KRT4 pre-mRNA was detected by qRT-PCR in HN6 cells. Results indicated that silence of METTL3 and METTL14 reduced structures of intron 2-exon 3 (KRT4-E(2)3), exon 3-intron 3 (KRT4-E(3)4) and exon 5-intron 5 (KRT4-E(5)6) in KRT4 pre-mRNA (Figs. 1A and 3C), which associated with junctions of exon 2-exon 3, exon 3-exon 4 and exon 5-exon 6 in KRT4 mRNA, respectively. By contrast, silence of METTL3 and METTL14 increased mature KRT4 mRNA level in HN6 cells (Fig. 3D). In addition, siRNA NC had no effect on levels of KRT4 pre-mRNA and mRNA (Figs. 3C and 3D).

Above results suggested that m6A methylation of exon-intron boundaries prevented intron splicing of KRT4 pre-mRNA in OSCC. Besides, these results also revealed that the m6A modification of m6A sites in exon 3 of KRT4 pre-mRNA nearby the exon-intron boundary (KRT4 pre-mRNA-E3(4)) could modify not only intron 2-exon 3 structure (KRT4-E(2)3) but also exon 3-intron 3 structure (KRT4-E(3)4). Therefore, silence of METTL3 and METTL14 by siRNAs could simultaneously reduce intron 2-exon 3 structure (KRT4-E(2)3) but also exon 3-intron 3 structure (KRT4-E(3)4).

m6A methylation inhibits the binding of DGCR8 to exon-intron boundaries in KRT4 pre-mRNA in OSCC

Bioinformatics analysis of CLIP-seq data from ENCORI database further found that splicing factor DGCR8 could bind to KRT4 mRNA in cancer cells (Fig. 4A). Furthermore, results of RIP demonstrated that DGCR8 associated with boundaries of intron 2-exon 3 (KRT4-E(2)3), exon 3-intron 3 (KRT4-E(3)4) and exon 5-intron 5 (KRT4-E(5)6) in KRT4 pre-mRNA in HN6 cells (Figs. 4B–4D), suggesting that DGCR8 might regulate KRT4 pre-mRNA splicing in HN6 cells. Moreover, silence of METTL3 and METTL14 facilitated the binding of DGCR8 to boundaries of intron 2-exon 3 (KRT4-E(2)3), exon 3-intron 3 (KRT4-E(3)4) and exon 5-intron 5 (KRT4-E(5)6) in KRT4 pre-mRNA (Figs. 4B–4D). All these data together suggested that m6A methylation suppressed the binding of DGCR8 to exon-intron boundaries in KRT4 pre-mRNA in OSCC.

Figure 4 m6A methylation inhibits the binding of DGCR8 to exon-intron boundaries in KRT4 pre-mRNA in OSCC.

(A) Potential binding site of DGCR8 in KRT4 mRNA. (B–D) Quantification of KRT4 pre-mRNA containing junctions of intron 2-exon 3 (KRT4-E(2)3, (B), exon 3-intron 3 (KRT4-E(3)4, (C) and exon 5-intron 5 (KRT4-E(5)6, (D) in KRT4 pre-mRNA by qRT-PCR following RIP performed by DGCR8 antibody in HN6 cells. E, exon. ∗P < 0.05, ∗∗P < 0.01, ∗∗∗P < 0.001.

Silence of DGCR8 prohibits intron splicing of KRT4 pre-mRNA in OSCC

Next, the role of DGCR8 in intron splicing of KRT4 pre-mRNA was demonstrated in HN6 cells. Results of qRT-PCR showed that silence of DGCR8 by siRNA (Fig. S1) increased structures of intron 2-exon 3 (KRT4-E(2)3), exon 3-intron 3 (KRT4-E(3)4) and exon 5-intron 5 (KRT4-E(5)6) in KRT4 pre-mRNA (Fig. 5A). In contrast, silence of DGCR8 decreased mature KRT4 mRNA level in HN6 cells (Fig. 5B). Besides, siRNA NC had no effect on levels of KRT4 pre-mRNA and mRNA (Figs. 5A and 5B). Thus, these results indicated that silence of DGCR8 prohibited intron splicing of KRT4 pre-mRNA in OSCC.

Figure 5 Silence of DGCR8 prohibits intron splicing of KRT4 pre-mRNA in OSCC.

(A) Quantification of KRT4 pre-mRNA containing junctions of intron 2-exon 3 (KRT4-E(2)3), exon 3-intron 3 (KRT4-E(3)4) and exon 5-intron 5 (KRT4-E(5)6) in KRT4 pre-mRNA by qRT-PCR in HN6 cells treated with or without DGCR8 siRNA. (B) Levels of KRT4 mRNA in HN6 cells treated with or without DGCR8 siRNA. NC, negative control; siDGCR8, DGCR8 siRNA; E, exon. ∗∗∗P < 0.001.

Discussion

The current study indicated that intron splicing of KRT4 pre-mRNA was suppressed in OSCC. Mechanistically, m6A methylation of exon-intron boundaries prevented intron splicing of KRT4 pre-mRNA in OSCC. Besides, m6A methylation suppressed the binding of DGCR8 to exon-intron boundaries in KRT4 pre-mRNA in OSCC, and silence of DGCR8 prohibited intron splicing of KRT4 pre-mRNA in OSCC.

Inhibition of intron splicing could lead to exon exclusion and subsequent expression of non-functional proteins. For instance, serine and arginine rich splicing factor 2 (SRSF2) prevents intron splicing to reduce exon 7 inclusion within survival of motor neuron (SMA) mRNA to produce non-functional SMA protein (Cho et al., 2015; Kashima et al., 2007; Moon et al., 2017). Besides, heterogenous ribonucleaoprotein C1 (hnRNP C1) facilitates exon inclusion within Ron mRNA through promoting intron 10 splicing (Moon et al., 2019). However, our recent study has indicated that KRT4 mRNA level is decreased in OSCC (Li et al., 2022). Therefore, inhibition of intron splicing in KRT4 pre-mRNA should not lead to expression of non-functional KRT4 protein in OSCC.

In addition, suppression of intron splicing also results in intron retention (Pendleton et al., 2017). Several studies have demonstrated that intron retention could result in nuclear pre-mRNA degradation. For example, intron retention stimulates nuclear methionine adenosyltransferase 2A (MAT2A) pre-mRNA decay under high S-adenosylmethionine (SAM) condition (Pendleton et al., 2017). Besides, poly(A)-binding protein nuclear 1 (PABPN1) protein negatively modifies its own expression through binding with PABPN1 pre-mRNA to enhance retention of the 3′-terminal intron and induce nuclear PABPN1 pre-mRNA degradation (Bergeron et al., 2015). Thus, inhibition of intron splicing in KRT4 pre-mRNA should result in intron retention and subsequent nuclear KRT4 pre-mRNA degradation in OSCC.

Growing evidence has indicated that m6A methylation plays a crucial role in intron retention. A previous study has revealed that METTL16 increases m6A level of a hairpin of MAT2A pre-mRNA to facilitate intron retention (Pendleton et al., 2017). By contrast, overexpression of alkB homolog 5 RNA demethylase (ALKBH5), which is a m6A-demethylase, enhances intron retention on E6 mRNA of human papillomavirus type 16 (Cui et al., 2022). Nevertheless, the mechanism of m6A methylation regulating intron retention on mRNA remains unclear.

A previous study has demonstrated that DGCR8 also regulates pre-mRNA splicing (Cirera-Salinas et al., 2017). Yet the role of DGCR8 in KRT4 pre-mRNA splicing has not been reported. Our results revealed that silence of DGCR8 prohibited intron splicing of KRT4 pre-mRNA in OSCC. Thus, this study uncovered the effect of DGCR8 on KRT4 pre-mRNA splicing for the first time.

DGCR8 also contributes to m6A methylation-mediated splicing of primary microRNAs (pri-miRNAs). In mammalian cells, METTL3 promotes m6A methylation of pri-miRNAs to enhance the binding of DGCR8 to pri-miRNAs and splicing of pri-miRNAs (Alarcon et al., 2015). By contrast, our data revealed that m6A methylation suppressed the binding of DGCR8 to exon-intron boundaries in KRT4 pre-mRNA to prohibit intron splicing of KRT4 pre-mRNA in OSCC. Therefore, DGCR8 might exert opposite effects on pre-mRNA splicing and miRNA processing.

A recent study has revealed that genes related to RNA methylation are associated with immunology, gene mutation and survival of OSCC patients (Wu, Tang & Cheng, 2022). Besides, METTL3 facilitates tumorigenesis and metastasis of OSCC by enhancing BMI1 m6A methylation (Liu et al., 2020). Moreover, METTL3 promotes OSCC progress by improving m6A methylation of protein arginine methyltransferase 5 (PRMT5) and programmed death-ligand 1 (PD-L1) (Ai et al., 2021). Therefore, above studies and our findings together suggest m6A methylation should facilitating OSCC progress.

The role of DGCR8 in OSCC has not been report. Nevertheless, two recent studies have indicated that DGCR8 enhances radiosensitive of head and neck squamous cell carcinoma (Long et al., 2021; Zhang et al., 2020b). Thus, previous studies and our results suggested that DGCR8 should play the opposite role of m6A methylation in OSCC. More importantly, these studies could further confirm the validity of our findings.

However, there were some limitations of the current study. For example, this study could be strengthened through identifying functional relevance of m6A methylation and DGCR8 to the suppression of OSCC cell growth induced by KRT4. Besides, the findings of this study should be confirmed by in vivo experiments.

Figure 6 Schematic diagram of molecular mechanisms for the current study.

The current study indicated that intron splicing of KRT4 pre-mRNA was suppressed in OSCC. Mechanistically, m6A methylation of exon-intron boundaries prevented intron splicing of KRT4 pre-mRNA in OSCC. In addition, m6A methylation suppressed the binding of DGCR8 to exon-intron boundaries in KRT4 pre-mRNA to prohibit intron splicing of KRT4 pre-mRNA in OSCC.

Conclusion

In summary, the current study indicated that intron splicing of KRT4 pre-mRNA was suppressed in OSCC. Mechanistically, m6A methylation of exon-intron boundaries prevented intron splicing of KRT4 pre-mRNA in OSCC. In addition, m6A methylation suppressed the binding of DGCR8 to exon-intron boundaries in KRT4 pre-mRNA to prohibit intron splicing of KRT4 pre-mRNA in OSCC (Fig. 6). These results revealed the mechanism downregulating KRT4 in OSCC and provided potential therapeutic targets for OSCC.

Supplemental Information

Supplemental Information 1 Raw data

Click here for additional data file.

Supplemental Information 2 Efficiencies of siRNAs

The level of METTL3 (A), METTL14 (A) and DGCR8 (B) detected by qRT-PCR in HN6 cells treated with or without siRNAs. NC, negative control; siMETTL3, METTL3 siRNA; siMETTL14, METTL14 siRNA; siDGCR8, DGCR8 siRNA. ∗∗∗P < 0.001, ∗∗∗∗P < 0.0001.

Click here for additional data file.

Additional Information and Declarations

Competing Interests

Author Contributions

Data Availability

The authors declare there are no competing interests.

Xiaoxu Li performed the experiments, prepared figures and/or tables, authored or reviewed drafts of the article, and approved the final draft.

Juan Fang performed the experiments, prepared figures and/or tables, and approved the final draft.

Xiaoan Tao performed the experiments, prepared figures and/or tables, and approved the final draft.

Juan Xia analyzed the data, prepared figures and/or tables, and approved the final draft.

Bin Cheng conceived and designed the experiments, analyzed the data, prepared figures and/or tables, authored or reviewed drafts of the article, and approved the final draft.

Yun Wang conceived and designed the experiments, authored or reviewed drafts of the article, and approved the final draft.

The following information was supplied regarding data availability:

The raw measurements are available in the Supplemental Files.

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
