# Peer review of "Splice site m6A methylation prevents binding of DGCR8 to suppress KRT4 pre-mRNA splicing in oral squamous cell carcinoma"

_PeerJ, doi:10.7717/peerj.14824_

## Round 0.1 · original submission · Minor Revisions

Dear Dr. Wang

Thank you for submitting your manuscript "plice site m6A methylation prevents binding of DGCR8 to suppress KRT4 pre-mRNA splicing in oral squamous cell carcinoma" to PeerJ. We have now sufficiently received reports from three reviewers who find the study interesting. Therefore, after careful consideration, we have decided to invite a minor revision of the manuscript.

As you will see from the reports copied below, the reviewers raised concerns regarding the functional relevance of m6A that require further validation to conclude the findings. Therefore, we ask you to address all of the reviewers' comments experimentally as appropriate. Without substantial revisions, we will be unlikely to send the paper back for review.

Important:
If you feel that you are able to comprehensively address the reviewers’ concerns, please provide a point-by-point response to these comments along with your revision. Please show all changes in the manuscript text file with track changes or color highlighting. If you are unable to address specific reviewer requests or find any points invalid, please explain why in the point-by-point response.

Best regards,

Abhishek Tyagi, PhD
Academic Editor

Reviewer 1 ·

Basic reporting

In the manuscript, the authors keep using splicing for miRNAs, in fact, the more appropriate term should be miRNA processing. For m6A-DGCR8-pri-miRNA processing, the original paper from Samuel Jafferrey's group should be cited.

Experimental design

No comment.

Validity of the findings

The conclusion is generally overstated. First, there is no evidence indicating the existence of m6A at the splice site. The resolution of MeRIP is not enough to conclude that. Second, no evidence to support that m6A prevents the binding of DGCR8 to KRT4 pre-mRNA; the siMettl3/14 results does not directly support m6A preventing DGCR8 binding. This is really important given that in miRNA cases the m6A helps to recruit DGCR8/DROSHA complex. Finally, the authors assumed that KRT4 transcription is not changed between different cell lines and under different treatments. However, this has to be validated by pol II ChIP. Finally, the introns without m6A modifications should be used as control.

Additional comments

The paper could be strengthened by testing functional relevance of m6A and DGCR8 to the suppression of cell growth by KRT4. In addition, the relevance of DGCR8 to splicing should be tested in NOK cells since KRT4 pre-mRNA has less m6A modification in these cells.

Reviewer 2 ·

Basic reporting

The present work made a great effort to address a critical research question. The authors reported that intron splicing of KRT4 pre-mRNA was suppressed in OSCC. Mechanistically, m6A methylation of exon-intron boundaries prevented intron splicing of KRT4 pre-mRNA in OSCC. Besides, m6A methylation suppressed the binding of splice factor DGCR8 microprocessor complex subunit (DGCR8) to exon-intron boundaries in KRT4 pre-mRNA to prohibit intron splicing of KRT4 pre-mRNA in OSCC. These findings revealed the mechanism downregulating KRT4 in OSCC and provided potential therapeutic targets for OSCC.

The work is well-planned and documented. Besides, the work is written in good English, outstanding presentation, and layout. Moreover, the research questions are valid, the language is clear and understandable, and the figures and the tables are vivid. In addition, all raw data have been shared.

Experimental design

The study design is appropriate to answer the research question and is detailed enough to be able to reproduce the study. Methods with sufficient information are also described.

Validity of the findings

The authors conducted a subjective discussion of the findings and compared the differences between previous studies and the current study, and the results supported their conclusions. Moreover, the conclusions are correct and cautious.

Additional comments

However, there are still deficiencies in the article, and I suggest that this manuscript should be processed with minor revision.

1. Why did the authors explore the mechanism regulating KRT4 pre-mRNA splicing in OSCC? They should state the reason in the “Instruction” section or “Results” section. 2. The term “mRNA splicing” is not correct, which should be revised to “pre-mRNA splicing” throughout the paper.
3. Some units of measurements in the “Materials and methods” section should be revised, such as “ml” should be revised to “mL”.
4. Effects of RNA methylation and DGCR8 on OSCC should be discussed. If RNA methylation and DGCR8 indeed contribute to OSCC progression, which could further confirm that the findings of the current study are valid.
5. Limitations of this study should be listed at the end of the “Discussion” section.
6. This manuscript is well written, yet some grammatical mistakes should be revised.

Reviewer 3 ·

Basic reporting

Overall, this manuscript is clearly written and logically structured, and the data are solid. Much appreciated by providing raw data.

Experimental design

In this manuscript, Xi et al., studied the role of m6A methylation and DGCR8 on KRT4 mRNA splicing in OSCC. They first profiled Intron splicing of KRT4 pre-mRNA and m6A methylation levels both in NOK cells and HN6 cells and investigated the correlation between m6A methylation and splicing efficiency. Inspired by the bioinformatics analysis, they further established the correlation between m6A methylation and the splicing factor DGCR8. By the overall silence of DGCR8 or METTL3 and METTL14, the two critical writers for m6A methylation, observed the intro splicing KRT4 pre-mRNA increase and decrease, respectively. This potentially indicates the mechanism of KRT4 down-regulation under OSCC.

Validity of the findings

1. The authors described those previous studies showed KRT4 is downregulated in OSCC, tissue samples, and patients…Are the authors tested the KRT4 levels in both cell lines they used in this study?
2. In the figure1, the authors observed a dramatic decrease in specific exon-exon junctions, especially the exon 2-exon 3 (E2/E3), and showed increased m6A levels of KRT4 pre-mRNA--E3(4) and E5(6). What is the m6A level of KRT4 pre-mRNA-E2(3)? And in Figure 3C, siRNA against METTL14 and METTL3 can also downregulate KRT4-E(2)3 level. Can the authors discuss this more?
3. The ability of DGCR8 to bind RNA is modulated by dimerization, in concert with acetylation and phosphorylation, reported by Wada et al.,2012…say, DGCR8 phosphorylation conditions are critical. Are the authors see the phosphorylation level changes along with the m6A methylation level?
4. siRNA knockdown efficiency is only confirmed by mRNA level, are the authors see protein level significantly downregulated?

Additional comments

5. There are several spelling errors, like line 51, line 53…

---

## Round 0.2 · accepted · Accept

Dear Dr. Wang,

We are delighted to accept your manuscript, entitled "Splice site m6A methylation prevents binding of DGCR8 to suppress KRT4 pre-mRNA splicing in oral squamous cell carcinoma," for publication in PeerJ.

Thank you for choosing to publish your interesting work with us.


With kind regards,
Abhishek Tyagi
Academic Editor, PeerJ

Reviewer 2 ·

Basic reporting

A satisfactory revision has been completed on this research work.
As requested, the authors have addressed all of my questions and revised the manuscript accordingly.
As a result, it is recommended to be published in its current form.

Experimental design

A satisfactory revision has been completed on this research work.
As requested, the authors have addressed all of my questions and revised the manuscript accordingly.
As a result, it is recommended to be published in its current form.

Validity of the findings

A satisfactory revision has been completed on this research work.
As requested, the authors have addressed all of my questions and revised the manuscript accordingly.
As a result, it is recommended to be published in its current form.

Additional comments

A satisfactory revision has been completed on this research work.
As requested, the authors have addressed all of my questions and revised the manuscript accordingly.
As a result, it is recommended to be published in its current form.

Reviewer 3 ·

Basic reporting

The authors has properly addressed my concerns.

Experimental design

The authors has properly addressed my concerns.

Validity of the findings

The authors has properly addressed my concerns.